# Mental health services accessibility for women and men: A protocol for a systematic review and meta-analysis

**Héllyda de Souza Bezerra**[1]*, **Ivani Iasmim de Araújo**[2], **Aryelly Dayane da Silva Nunes**[2], **Isabelle Ribeiro Barbosa**[3]

**1** Graduate Program in Public Health, Federal University of Rio Grande do Norte (UFRN), Natal, Rio Grande do Norte, Brazil, **2** Federal University of Rio Grande do Norte (UFRN), Natal, Rio Grande do Norte, Brazil, **3** Faculty of Health Sciences of Trairi (FACISA), Federal University of Rio Grande do Norte (UFRN), Natal, Rio Grande do Norte, Brazil

* hellydasbezerra@hotmail.com

**Data Availability Statement:** No datasets were generated or analysed during the current study. All relevant data from this study will be made available upon study completion.

## Abstract

### Introduction

Mental and behavioral disorders constitute a serious public health problem and require adequate access among women and men for promotion, prevention and treatment of mental illness.

### Methods and analysis

For the writing of this protocol we will use the guidelines of the PRISMA-P Checklist (Main Items for Reporting Systematic Reviews and Meta-analyzes). This protocol was registered under the number: CRD42021243263. To this end, research will be conducted in the PubMed, Web of Science, Scopus, CINAHL and ScienceDirect databases in search of cross-sectional studies that assess the prevalence of access to mental health services among women and men. All cross-sectional studies that examined the prevalence of mental health services accessibility among women and men will be included. The search will be conducted by two independent researchers who will identify the articles; they will exclude duplicate studies. Through a blinded assessment, they will select articles using the Rayyan QCRI application. The methodological quality of the included studies will be assessed by the Joanna Briggs Institute Checklist for Analytical Cross-Sectional Studies. Meta-analyses will be performed according to the conditions of the included data.

### Ethics and disclosure

For the development of this study, there is no need for ethical review, as this is a systematic review that will use secondary studies. The conclusions of this study will be disseminated through peer-reviewed publications, conference presentations and condensed abstracts to key stakeholders and partners in the field. The database search is scheduled to start on May 10th, 2021. The entire review process is expected to be completed by August 30th, 2021.

**Funding:** The author(s) received no specific funding for this work.

**Competing interests:** The authors have declared that no competing interests exist.

## Introduction

Mental disorders represent 12% of the total burden of diseases, being configured as a serious public health problem due to their high prevalence and serious effects on the quality of life in all collective spheres. Studies indicate that one in ten people may have some mental disorder throughout their lives; in addition, this occurrence can be aggravated by several social factors and inequities [1, 2].

Factors such as socioeconomic status, age and gender directly influence the prevalence of mental illness. Research highlights that depression and anxiety disorders affect more women than men in a global estimate. Such discussion has been gaining prominence due to the high rates of suicide, being the cause of death of 788,000 people in the year 2015 [3].

Since there is a difference in the prevalence of mental and behavioral disorders between men and women, the question of if there is a difference in accessibility to mental health services according to gender comes up.

Starfield [4] refers to accessibility as the possibility people have or not to reach health services. Access, on the other hand, is given as the way people experience it. Pedraza et al. [5] introduces accessibility as the ease in using the health services and the readjustment between the characteristics of the resources and the population, while Barra et al. [6] brings accessibility associated with various factors such as availability of care, location of the facility, besides the communication barriers between teams and users.

In mental health services, its access is influenced by factors such as age, socioeconomic status, race and gender. As for the latter, it follows a historical burden, where health care has always focused on women and children, while men have a perpetuation of gender stereotypes of strength and virility, not being part of their socialization of self-care, causing the neglect of clinical signs and symptoms, as well as the lack of demand for mental health services [7].

With regard to women, there is an overburden that causes more illness, such as reconciling work and domestic responsibilities, labor market demands, wage devaluation, high violence rates, among others, that lead to a greater demand for medical support [8].

It is necessary to understand that men and women experience the process of mental illness and the way they seek help in a differentiated manner. This may be related to the diverse historical and cultural backgrounds. The recognition of this subjectivity and individual needs should enable the construction of public policies for mental health that ensure the principle of equality, thus minimizing disparities in mental health services accessibility.

Accordingly, from the guiding question of the study: is there a difference in mental health services accessibility in relation to women and men? The objective of the systematic review is to analyze the prevalence of access to mental health services according to women and men.

## Methods

### Protocol and register

The protocol is being reported and directed in accordance with the Preferred Reporting Items for Systematic Review and Meta-Analysis Protocols (PRISMA-P) Statement [9].

This systematic review was registered in the International Prospective Register of Systematic Reviews (PROSPERO) on Apr 16th, under protocol CRD42021243263 Available at: https://www.crd.york.ac.uk/prospero/display_record.php?RecordID=243263

### Electronic searches

At first, the identification of articles will be performed in the electronic databases: PubMed, CINAHL, Web of Science, Scopus and Science Direct. If necessary, the search strategy will be

modified for each database in order to perform the appropriate search. Two reviewers will perform a double-blind search to identify eligible studies.

The pair of independent researchers will perform the search, and publications considered potentially relevant will be included in the review if they meet all the inclusion criteria. Consensus meetings will be held at each step; if necessary, a third reviewer will participate in them.

The reference list of possible included studies will be screened to identify other relevant publications. In case of disagreement, it will be resolved by a third reviewer.

Fig 1 displays the flowchart adapted from PRISMA-P16 [9] containing all steps of study selection for this review.

## Search strategy

The search strategy is displayed in Table 1.

## Inclusion criteria

For this review, articles meeting the eligibility criteria based on Population, exposure, Comparison, Outcome and Study Design (PECOS) of the study will be included, as described in Table 2.

Studies that present the following information will be included:

i. adult women and men (over 18 years old)

ii. Mental health services accessibility

iii. The prevalence mental health services accessibility between men and women

iv. Cross-sectional studies

v. No language, year or country restrictions

Access is the opportunity to use health services when needed, expressing characteristics of its offer and circumstances that facilitate or hinder people's ability to use it effectively. Access is simultaneously related to four elements: availability, accessibility, acceptability, and quality.

Primary, secondary or tertiary care units will be considered as health units, since in Brazil mental health care takes place at the three levels of care. Information about age and gender may be self-reported or may come from medical records.

## Exclusion criteria

Articles whose studies include people less than 18 years of age and pregnant women, studies with accessibility to services other than mental health; studies with women and/or men who identify themselves by non-biological gender characteristics and Studies whose prevalence calculation with raw data is not possible. Cohort and case-control studies, case reports, randomized clinical trials and reviews, as well as qualitative studies, will also be excluded.

## Study selection

For the selection of the studies based on the inclusion criteria, the following steps will be performed: 1) exclusion of duplicate articles, 2) reading of the title and abstract of all articles; and, finally, 3) reading of the studies selected in the previous step. The Rayyan QCRI [10] software will be used to perform these steps.

The selection phase of the studies will be conducted by two independent researchers; and, in case of disagreement between the researchers, even after the consensus meeting, the third researcher will be involved.

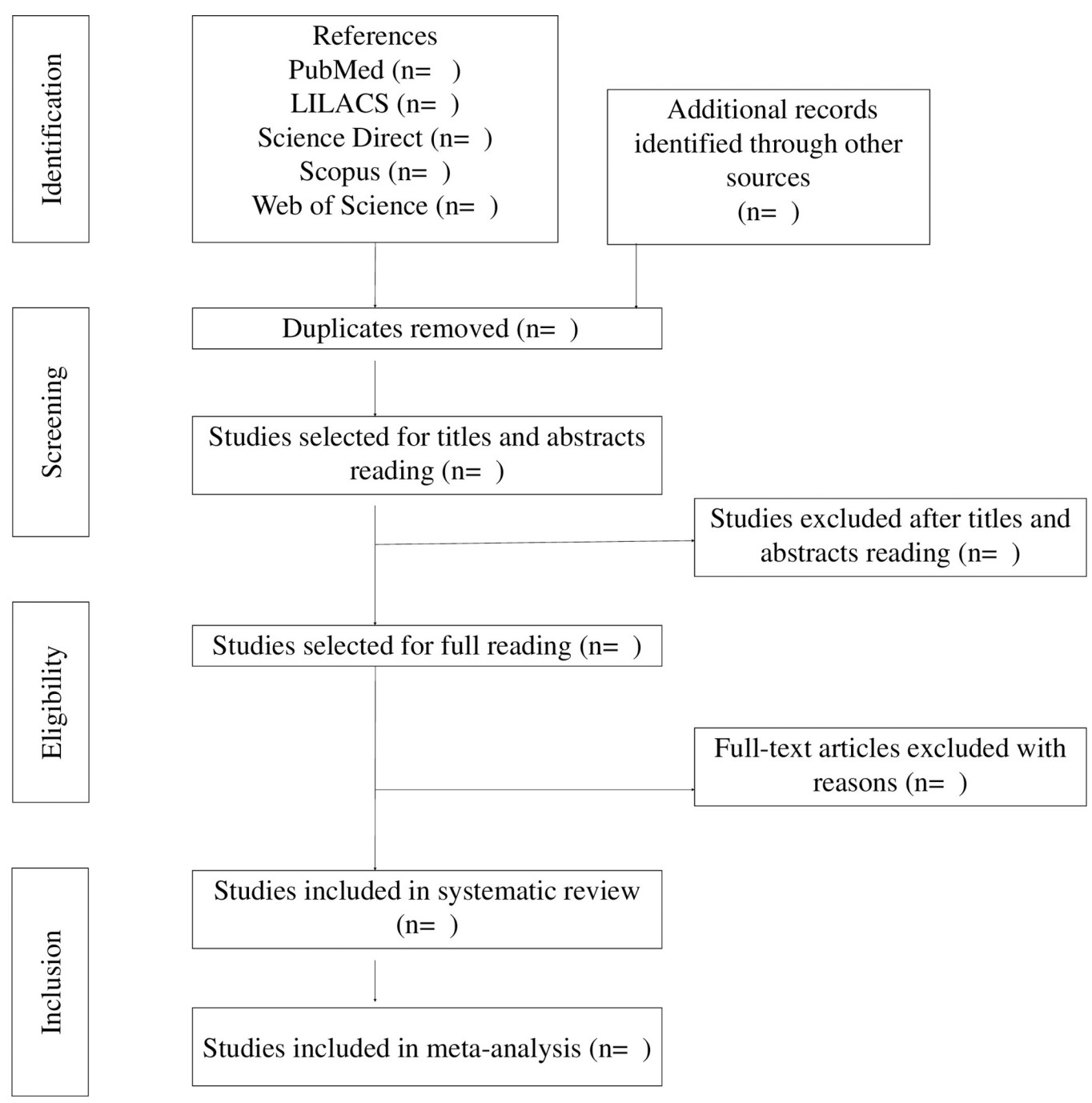

**Fig 1. Flowchart.** Adapted from PRISMA-P.

Mendeley Software will be used to format the references [11].

## Data extraction

Study characteristics (author, publication date, study design, time period and study site) and study population (gender and age range of participants) will be extracted from all included studies. Sample size information will also be included, to consider articles that include a

**Table 1. Search strategy.**

| Search Items | Keywords |
|---|---|
| 1. | Health Services Accessibility |
| 2. | Health Services Accessibility |
| 3. | Universal Access to Health Care Services |
| 4. | Access to Health Services |
| 5. | Access to Health Care |
| 6. | Equity in Access to Health Services |
| **7.** | **OR 1/6** |
| 8. | Mental Health Care |
| 9. | **Community Mental Health Services** |
| 10. | Mental Health Services |
| 11. | Psychosocial Care Center |
| 12. | Mental Hygiene Service |
| 13. | Mental Health Care Service |
| 14. | Psychiatric Service |
| **15.** | **OR 8/14** |
| 16. | Women |
| 17. | Woman |
| 18. | Women's Groups |
| 19. | Women's Groups |
| 20. | Women's Groups |
| **21** | **OR 16/20** |
| 22. | Men |
| **23.** | **OR 22/22** |
| **24.** | **7 AND 15 AND 21 AND 23** |

representative sample. Peer-reviewed publications that include the following criteria will be identified: adult women and men (participants); mental health services accessibility (exposure). The outcome measure will be the prevalence of access to mental health services among women and men. When there is no access prevalence result and there are data for the calculation, the prevalence will be calculated by the authors. Review Manager software (RevMan 2010) will be used to perform statistical analysis.

## Risk of bias and quality assessment

The methodological quality of the included studies will be assessed by the Joanna Briggs Institute Checklist for Analytical Cross-Sectional Studies [12].

The eight questions will be answered by the first and second reviewers independently with "Yes", "No", "Unclear", or "Not applicable". When necessary, a consensus meeting will be held. The risk of bias results will be classified into (1) low risk, if studies achieve more than

**Table 2. PECO description.**

| Abbreviation | PECO | Elements |
|---|---|---|
| P | Participants | Adult women |
| E | Exposition | Accessibility to mental health services |
| C | Comparison | inaccessibility to mental health services |
| O | Outcome | Prevalence of access to mental health services |

70% "yes" scores; (2) moderate risk, if studies achieve between 50% and 69% "yes" scores; and (3) high risk, if studies achieve less than 49% "yes" scores [13]. The graphic illustration of the risk of bias will be created using Review Manager 5.3 software (RevMan 5.3, The Nordic Cochrane Centre, Copenhagen, Denmark).

## Data synthesis

Results will be expressed as prevalence with 95% confidence intervals (CI). Random effects models will be chosen depending on if there is an absence or presence of heterogeneity among studies. The I2 statistic will assess statistical heterogeneity (<25%, no heterogeneity; 25% -50%, moderate heterogeneity; and > 50%, strong heterogeneity).

When there is significant heterogeneity among the included studies (I2> 50%), a random effects model will be used for the analysis; otherwise, the fixed effects model will be used.

All tests will be performed by means of Review Manager software (RevMan version 5.3.0) and a two-tailed p value <0.05 will be considered statistically significant.

Data underlying the findings of the systematic review will be provided as part of the manuscript or deposited in a public repository. There are no results in this study, and that there no other articles submitted or published with the results of this research.

## Reliance on cumulative evidence

The GRADE approach will be used to assess the quality of the evidence that will be included in this review.

## Ethics and disclosure

The study will be conducted following this protocol, which was approved by PROSPERO in April 2021: The database search will begin on April 30th, 2021, and the entire review process is expected to be completed by August 30th, 2021. The results will be published in peer-reviewed journals and local, national, and international conference proceedings.

## Patient and public involvement

No patients will be involved.

## Parcial results

Fig 2 will show the articles selected from PRISMA.

# Discussion

It is understood that men and women have needs beyond their stereotypes. Therefore, it is extremely necessary to modify the positioning of prevention and health promotion actions, as well as the forms of intervention as health professionals. Gender issues need to be understood and carried out as a transversal content to all health topics, and not linked to groups or specific campaigns. The invisibility of gender issues has been leading to the exclusion from health care services and, consequently, to illness and lethality [7]. The difference in mental health services accessibility according to men and women can bring consequences for the prevention of injuries, health promotion and treatment for mental health services, thus hindering adherence and continuity of treatment for mental disorders.

As for the limitations of the included studies, as they are cross-sectional studies, they do not estimate the causality of the studied variables. Still the dependence on the quality and adequate reporting of primary studies, as well as the difficulty in combining studies that consider

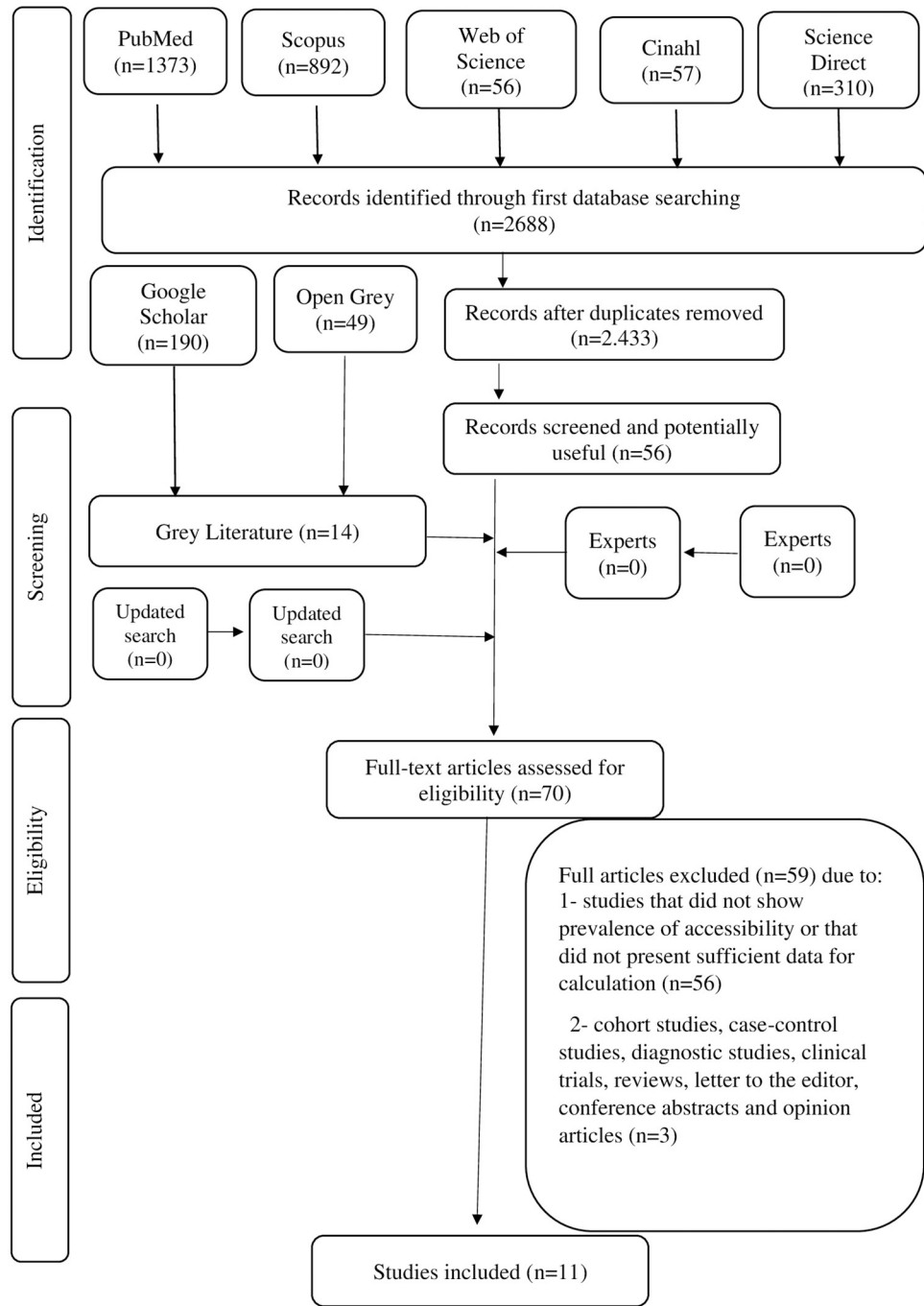

**Fig 2. Flowchart of article selection adapted from PRISMA.** Difference in mental health services accessibility for women and men: a systematic review. Brazil, 2021.

different meanings for accessibility to health services. In addition, information from studies is self-reported, which can generate information bias.

Therefore, this systematic review will help enhance existing public mental health policies and contribute to the development of new strategies that can ensure a more effective and equal accessibility to mental health services, regardless of gender. Accordingly, it is expected that the

conclusion of this systematic review will produce results that make it possible to understand that adequate accessibility contributes to the improvement of the treatment process offered to men and women who need to use mental health services. Identifying if there are differences in the accessibility of mental health services between women and men can help reduce the effects of gender inequalities, so that they can be equal to the diverse individuals.

## Supporting information

**S1 Checklist. PRISMA-P (Preferred Reporting Items for Systematic review and Meta-Analysis Protocols) 2015 checklist: Recommended items to address in a systematic review protocol**[*]**.**
(DOC)

## Author Contributions

**Conceptualization:** Héllyda de Souza Bezerra, Ivani Iasmim de Araújo, Aryelly Dayane da Silva Nunes, Isabelle Ribeiro Barbosa.

**Data curation:** Héllyda de Souza Bezerra, Ivani Iasmim de Araújo.

**Investigation:** Héllyda de Souza Bezerra, Ivani Iasmim de Araújo, Aryelly Dayane da Silva Nunes, Isabelle Ribeiro Barbosa.

**Methodology:** Héllyda de Souza Bezerra, Ivani Iasmim de Araújo, Aryelly Dayane da Silva Nunes, Isabelle Ribeiro Barbosa.

**Project administration:** Héllyda de Souza Bezerra, Ivani Iasmim de Araújo, Aryelly Dayane da Silva Nunes, Isabelle Ribeiro Barbosa.

**Software:** Héllyda de Souza Bezerra.

**Supervision:** Héllyda de Souza Bezerra, Ivani Iasmim de Araújo, Aryelly Dayane da Silva Nunes, Isabelle Ribeiro Barbosa.

**Visualization:** Héllyda de Souza Bezerra, Ivani Iasmim de Araújo, Aryelly Dayane da Silva Nunes, Isabelle Ribeiro Barbosa.

**Writing – original draft:** Héllyda de Souza Bezerra, Ivani Iasmim de Araújo, Aryelly Dayane da Silva Nunes, Isabelle Ribeiro Barbosa.

**Writing – review & editing:** Héllyda de Souza Bezerra, Ivani Iasmim de Araújo, Aryelly Dayane da Silva Nunes, Isabelle Ribeiro Barbosa.

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
