## [Decision Letter · Decision Letter 0]

3 Jun 2022

PONE-D-21-14262MENTAL HEALTH SERVICES ACCESSIBILITY FOR WOMEN AND MEN: A PROTOCOL FOR A SYSTEMATIC REVIEW AND META-ANALYSISPLOS ONE

Dear Dr. Bezerra,

Thank you for submitting your manuscript to PLOS ONE. After careful consideration, we feel that it has merit but does not fully meet PLOS ONE’s publication criteria as it currently stands. Therefore, we invite you to submit a revised version of the manuscript that addresses the points raised during the review process.

Please, review the protocol based on the reviewer 1 comments and consider the following aspects. 

The author cited the PRISMA Statement as methodological guidance; however, the PRISMA is considered a reporting guideline. Please, see the following editorial for more information: https://systematicreviewsjournal.biomedcentral.com/articles/10.1186/s13643-021-01671-z

The decision to use fixed and random effects in the meta-analysis should be reviewed because using I2 statistics to do that is not recommended. 

We look forward to receiving your revised manuscript.

Kind regards,

Rafael Sarkis-Onofre

Academic Editor

PLOS ONE

Journal Requirements:

Additional Editor Comments (if provided):

Academic Editor:

Please, review the protocol based on the reviewer 1 comments and consider the following aspects.

The author cited the PRISMA Statement as methodological guidance; however, the PRISMA is considered a reporting guideline. Please, see the following editorial for more information: https://systematicreviewsjournal.biomedcentral.com/articles/10.1186/s13643-021-01671-z

The decision to use fixed and random effects in the meta-analysis should be reviewed because using I2 statistics to do that is not recommended.

Reviewers' comments:

Reviewer's Responses to Questions

**Comments to the Author**

1. Does the manuscript provide a valid rationale for the proposed study, with clearly identified and justified research questions?

Reviewer #1: Yes

2. Is the protocol technically sound and planned in a manner that will lead to a meaningful outcome and allow testing the stated hypotheses?

Reviewer #1: Partly

3. Is the methodology feasible and described in sufficient detail to allow the work to be replicable?

Reviewer #1: No

4. Have the authors described where all data underlying the findings will be made available when the study is complete?

Reviewer #1: No

5. Is the manuscript presented in an intelligible fashion and written in standard English?

Reviewer #1: Yes

6. Review Comments to the Author

You may also provide optional suggestions and comments to authors that they might find helpful in planning their study.

Reviewer #1: Dear authors,

The research protocol presents a very relevant theme. The study is justified based on the information presented in the protocol. The idea is interesting, but the study needs adjustments and improvements before being considered for publication and generating significant results.

Some points must be considered:

1. Considering the dates presented in the protocol for the beginning of the searches and the conclusion of the reviews, we can assume that the project already presents results, so why not present them?

2. The registration date in PROSPERO presented on the platform is 16 April 2021 and not 10 May as informed in the methodology.

3. There appears to be misuse of the PRISMA-P statement. Misuse often appears through authors trying to use the PRISMA statement and its extensions as a methodological guideline. Phrases such as "The project will be CONDUCTED/PERFORMED/DEVELOPED based on PRISMA-P." to justify the use of PRISMA are wrong.

4. There is inconsistency regarding the databases that will be consulted. The "CINAHL" database is mentioned in the abstract and in the "Electronic surveys" topic, but it does not appear in the flowchart, where the "LILACS" database is mentioned.

5. Table 1 referenced in the text (search strategy) does not match table 1 presented (PICO description). The same goes for table 2.

6. In the case of a systematic review of cross-sectional observational studies, the acronym PICO needs to be reviewed, as well as its description throughout the text and in the table.

7. There is a lack of a more detailed description in the study methodology, mainly in the eligibility criteria. This is critical for the reproducibility of the study.

8. Different accessibility concepts and approaches are mentioned in the introduction. How will the authors take care of this in the inclusion and analysis of the different studies in the review?

9. Are there any restrictions during the search or selection of studies (language restrictions, for example)?

10. How will authors deal with missing data in data extraction and evaluation of included studies?

11. Justify why you used the mentioned classification in topic "Risk of bias assessment" for the results of the Joanna Briggs Institute for Analytical Cross-Cutting Studies Checklist.

12. Add possible study limitations at the end of the discussion.

13. Authors need to describe where all data underlying the findings will be made available when the study is complete.

7. PLOS authors have the option to publish the peer review history of their article (what does this mean?). If published, this will include your full peer review and any attached files.

Reviewer #1: No

---

## [Author Response · Author response to Decision Letter 0]

7 Jul 2022

Dear Reviewer,

We appreciate contributions to our protocol article. We hope that this article is accepted and that it can contribute to your journal that has a great international impact. All points suggested were accepted and corrected in the article file.

The following points were corrected and answered:

• 1- The protocol registration date was corrected;

• 2- As the systematic review project was already underway, only a partial result was included in the protocol article, as the systematic review article is yet to be submitted. If partial data is not necessary, please inform that they will be removed from the protocol article.

• 3- The sentence was corrected using PRISMA for the protocol. The issue of using PRISMA as methodological guidance was modified in the methodology, we modified the sentence to "The protocol is being reported and directed according to the Declaration of Preferred Reporting Items for Systematic Review and Meta-Analysis Protocols (PRISMA-P) " according to the suggested methodology: Sarkis-Onofre, R., Catalá-López, F., Aromataris, E. et al. How to properly use the PRISMA Statement. System Rev 10, 117 (2021). https://doi.org/10.1186/s13643-021-01671-z

• 4- The LILACS database was modified in the figure of the flowchart, since the database used was CINAHL;

• 5- The descriptions in tables 1 and 2 have been corrected;

• 6- As the protocol involves observational studies, the acronym PICO was changed to PECO, in which the studies involve exposure and non-intervention;

• 7- A more detailed description of the study eligibility criteria has been included;

• 8- Regarding the concepts of accessibility questioned in the introduction, accessibility is a very broad concept with several aspects, taking into account this, it was worked on in the systematic review as access in general, resulting in the arrival of the woman or the man in the mental health service;

• 9- There were no language, country or year restrictions regarding the studies included in the protocol/systematic review. This information was adjusted in the methodology;

• 10- Regarding missing data in data extraction, when the data on the prevalence of accessibility to mental health services is absent, if they have data available for the calculation, the authors will perform the prevalence calculation. Regarding the evaluation of the included studies, studies with a representative sample will be considered. Studies will be assessed for bias by the Jonna Brigs checklist for cross-sectional studies;

• 11- The term has been changed to "Risk of bias and quality assessment" as it better reproduces the objective of the Joanna Briggs assessment, which includes an assessment of the methodological quality of a study and determining the extent to which a study addressed the possibility of bias in its design, conduct and analysis;

• 12- They were included as a study in the discussion.

---

## [Decision Letter · Decision Letter 1]

19 Aug 2022

PONE-D-21-14262R1MENTAL HEALTH SERVICES ACCESSIBILITY FOR WOMEN AND MEN: A PROTOCOL FOR A SYSTEMATIC REVIEW AND META-ANALYSISPLOS ONE

Dear Dr. Bezerra,

Thank you for submitting your manuscript to PLOS ONE. After careful consideration, we feel that it has merit but does not fully meet PLOS ONE’s publication criteria as it currently stands. Therefore, we invite you to submit a revised version of the manuscript that addresses the points raised during the review process.

We look forward to receiving your revised manuscript.

Kind regards,

Rafael Sarkis-Onofre

Academic Editor

PLOS ONE

Additional Editor Comments (if provided):

The reviewer highlighted that some important aspects were not reviewed. Please, review the protocol accordingly.

Reviewers' comments:

Reviewer's Responses to Questions

**Comments to the Author**

1. Does the manuscript provide a valid rationale for the proposed study, with clearly identified and justified research questions?

Reviewer #1: Yes

2. Is the protocol technically sound and planned in a manner that will lead to a meaningful outcome and allow testing the stated hypotheses?

Reviewer #1: Yes

3. Is the methodology feasible and described in sufficient detail to allow the work to be replicable?

Reviewer #1: No

4. Have the authors described where all data underlying the findings will be made available when the study is complete?

Reviewer #1: Yes

5. Is the manuscript presented in an intelligible fashion and written in standard English?

Reviewer #1: Yes

6. Review Comments to the Author

You may also provide optional suggestions and comments to authors that they might find helpful in planning their study.

Reviewer #1: Dear authors,

Despite noting the improvements with the revisions carried out, I still consider that the study needs adjustments before being considered for publication and generating good results. Some points must be considered:

1. Study protocols should describe detailed plans and proposals for research projects. Also, according to the journal's submission guide (https://journals.plos.org/plosone/s/submission-guidelines#loc-study-protocols), the protocol must: 1. Relate to a research study that HAS NOT YET GENERATED RESULTS; 2. Be SUBMITTED BEFORE participant recruitment or DATA COLLECTION for the study is complete. Considering the projection of the beginning and end of the review process presented in the protocol, as well as the presence of preliminary results, it does not make sense to publish this protocol.

2. In the abstract, the erroneous statement that the protocol will be DEVELOPED based on the PRISMA-P guideline continues. PRISMA and its extensions only guide reporting, not the conduct of studies and protocols.

3. The phrase "The protocol is being reported and directed in accordance with the Preferred Reporting Items for Systematic Review and Meta-Analysis Protocols (PRISMA-P)" should be transferred to the topic "Protocol and register" as it has no relation to "Electronic searches"

4. Regarding the description of the reported acronym PECO (Participants: Adult women; Exposition: Accessibility to mental health services; Comparison: Adult men; Outcome: Prevalence of access to mental health services). Comparison should be the control for exposure, in my view, inaccessibility to mental health services. Description must be revised.

5. Although the review provides a more detailed description of the eligibility criteria, these and the methodology as a whole still lack details. For example, what the authors consider accessibility to health services can be described in the inclusion criteria (as already answered in question 8). What will be considered a mental health service? Will gender be collected and analyzed in the study only when it was reported by the patient, by data from medical records, or both?

6. In the exclusion criteria in the methodology, it is reported that studies that do not show prevalence of accessibility will be excluded. However, in the data extraction it is mentioned that even when these data are absent, studies will be considered, if the calculation of prevalence with raw data is possible. Need standardization!

7. Limitations such as dependence on the quality and adequate reporting of primary studies, as well as the difficulty in combining studies that consider different meanings for accessibility to health services, were not considered.

7. PLOS authors have the option to publish the peer review history of their article (what does this mean?). If published, this will include your full peer review and any attached files.

Reviewer #1: No

---

## [Author Response · Author response to Decision Letter 1]

3 Oct 2022

Dear Reviewer,

We appreciate contributions to our protocol article. We hope that this article is accepted and that it can contribute to your journal that has a great international impact. All points suggested were accepted and corrected in the article file.

The following points were corrected and answered:

• 1- we have included a sentence in the methods that indicates that there are no results in this study, and that there no other articles submitted or published with the results of this research;

• 2- The statement in the abstract was corrected.

• 3- The phrase "The protocol is being reported and directed in accordance with the Preferred Reporting Items for Systematic Review and Meta-Analysis Protocols (PRISMA-P)" was transferred to the topic "Protocol and register".

• 4- The description of table 2 was corrected.;

• 5- the requested answers were included in the methods section, in the item "inclusion criteria";

• 6-The exclusion criteria have been corrected;

• 7- The suggested limitations have been added.

---

## [Editor Report · Decision Letter 2]

2 Nov 2022

MENTAL HEALTH SERVICES ACCESSIBILITY FOR WOMEN AND MEN: A PROTOCOL FOR A SYSTEMATIC REVIEW AND META-ANALYSIS

PONE-D-21-14262R2

Dear Dr. Bezerra,

We’re pleased to inform you that your manuscript has been judged scientifically suitable for publication and will be formally accepted for publication once it meets all outstanding technical requirements.

Kind regards,

Rafael Sarkis-Onofre

Academic Editor

PLOS ONE

Additional Editor Comments (optional):

Academic Editor:

All of my concerns were addressed.
---

## [Editor Report · Acceptance letter]

7 Nov 2022

PONE-D-21-14262R2 

MENTAL HEALTH SERVICES ACCESSIBILITY FOR WOMEN AND MEN: A PROTOCOL FOR A SYSTEMATIC REVIEW AND META-ANALYSIS 

Dear Dr. Bezerra:

I'm pleased to inform you that your manuscript has been deemed suitable for publication in PLOS ONE. Congratulations! Your manuscript is now with our production department. 

Kind regards, 

on behalf of

Dr. Rafael Sarkis-Onofre 

Academic Editor

PLOS ONE